# CLASSIFICATION-DENOISING NETWORKS

## ABSTRACT

Image classification and denoising suffer from complementary issues of lack of robustness or partially ignoring conditioning information. We argue that they can be alleviated by unifying both tasks through a model of the joint probability of (noisy) images and class labels. Classification is performed with a forward pass followed by conditioning. Using the Tweedie-Miyasawa formula, we evaluate the denoising function with the score, which can be computed by marginalization and backpropagation. The training objective is then a combination of cross-entropy loss and denoising score matching loss integrated over noise levels. Numerical experiments on CIFAR-10 and ImageNet show competitive classification and denoising performance compared to reference deep convolutional classifiers/denoisers, and significantly improves efficiency compared to previous joint approaches. Our model shows an increased robustness to adversarial perturbations compared to a standard discriminative classifier, and allows for a novel interpretation of adversarial gradients as a difference of denoisers. [1]

## 1 INTRODUCTION: CLASSIFICATION AND DENOISING

Image classification and denoising are two cornerstone problems in computer vision and signal processing. The former aims to associate a label $c \in \{1, \ldots, C\}$ to input images $\mathbf{x}$. The latter involves recovering the image $\mathbf{x}$ from a noisy observation $\mathbf{y} = \mathbf{x} + \sigma\boldsymbol{\epsilon}$. Before the 2010s, these two problems were addressed with different kinds of methods but similar mathematical tools like wavelet decompositions, non-linear filtering and Bayesian modeling. Since the publication of AlexNet (Krizhevsky et al., 2012), deep learning (LeCun et al., 2015) and convolutional neural network (CNNs) (LeCun et al., 1998) have revolutionized both fields. ResNets (He et al., 2016) and their recent architectural and training improvements (Liu et al., 2022; Wightman et al., 2021) still offer close to state-of-the-art accuracy, comparable with more recent methods like vision transformers (Dosovitskiy et al., 2020) and visual state space models (Liu et al., 2024). In image denoising, CNNs lead to significant improvements with the seminal work DnCNN (Zhang et al., 2017b) followed by FFDNet (Zhang et al., 2018) and DRUNet (Zhang et al., 2021). Importantly, deep image denoisers are now at the core of deep generative models estimating the gradient of the distribution of natural images (Song & Ermon, 2019), a.k.a. score-based diffusion models (Ho et al., 2020).

However, some unsolved challenges remain in both tasks. For one, classifiers tend to interpolate their training set, even when the class labels are random (Zhang et al., 2016), and are thus prone to overfitting. They also suffer from robustness issues such as adversarial attacks (Szegedy, 2013). Conversely, deep denoisers seem to neither overfit nor memorize their training set when it is sufficiently large (Yoon et al., 2023; Kadkhodaie et al., 2023). On the other hand, when used to generate images conditionally to a class label or a text caption, denoisers are known to occasionally ignore part of their conditioning information (Conwell & Ullman, 2022; Rassin et al., 2022), requiring ad-hoc techniques like classifier-free guidance (Ho & Salimans, 2022) or specific architecture and synthesis modifications (Chefer et al., 2023; Rassin et al., 2024). Another issue is that optimal denoisers should be conservative vector fields but DNN denoisers are only approximately conservative (Mohan et al., 2020), and enforcing this property is challenging (Saremi, 2019; Chao et al., 2023).

We introduce a conceptual framework which has the potential to address these issues simultaneously. We propose to learn a single model parameterizing the *joint* log-probability $\log p(\mathbf{y}, c)$ of noisy images $\mathbf{y}$ and classes $c$. Both tasks are tackled with this common approach, aiming to combine their

---

[1]We will release code upon acceptance.

strengths while alleviating their weaknesses. First, the model gives easy access to the conditional log-probability $\log p(c|\mathbf{y})$ by conditioning, allowing to classify images with a single forward pass. Second, we can obtain the marginal log-probability $\log p(\mathbf{y})$ by marginalizing over classes $c$, and compute its gradient with respect to the input $\mathbf{y}$ with a backward pass. The denoised image can then be estimated using the Tweedie-Miyasawa identity (Robbins, 1956; Miyasawa et al., 1961; Raphan & Simoncelli, 2011). We can also perform class-conditional denoising similarly using $\nabla_{\mathbf{y}} \log p(\mathbf{y}|c)$.

Previous related works in learning joint energy-based models (Grathwohl et al., 2019) or gradient-based denoisers (Cohen et al., 2021; Hurault et al., 2021; Yadin et al., 2024) have faced two key questions, respectively concerning the training objective and network architecture. Indeed, learning a probability density over high-dimensional images is a very challenging problem due to the need to estimate normalization constants, which has only been recently empirically solved with score-matching approaches (Song et al., 2021). Additionally, different architectures, and therefore inductive biases, are used for both tasks: CNN classifiers typically have a feedforward architecture which maps an image $\mathbf{x}$ to a logits vector $(\log p(c|\mathbf{x}))_{1 \le c \le C}$, while CNN denoisers have a UNet encoder/decoder architecture which outputs a denoised image $\hat{\mathbf{x}}$ with the same shape as the input image $\mathbf{x}$. The joint approach requires unifying these two architectures in a single one whose *forward* pass corresponds to a classifier and *forward plus backward* pass corresponds to a denoiser, while preserving the inductive biases that are known to work well for the two separate tasks.

Here, we propose principled solutions to these questions. As a result of our unifying conceptual framework, we derive a new interpretation of adversarial classifier gradients as a difference of denoisers, which complements previous connections between adversarial robustness and denoising. Our approach also opens new research directions: having direct access to a log-probability density $\log p(\mathbf{y}, c)$ can be expected to lead to new applications, such as out-of-distribution detection or improved interpretability compared to score-based models.

The contributions of this paper are the following:

- We introduce a framework to perform classification, class-conditional and unconditional denoising with a single network parameterizing the joint distribution $p(\mathbf{y}, c)$. The two training objectives naturally combine in a lower-bound on the likelihood of the joint model. Further, our approach evidences a deep connection between adversarial classifier gradients and (conditional) denoising. Pursuing this line of research thus has the potential to improve both classifier robustness and denoiser conditioning.

- We propose an architecture to parameterize the joint log-probability density of images and labels which we call GradResNet. It makes minimal modifications to a ResNet architecture to incorporate inductive biases from UNet architectures appropriate to denoising (when computing a backward pass), while preserving those for classification (in the forward pass).

- We validate the potential of our method on the CIFAR-10 and ImageNet datasets. In particular, our method is significantly more efficient and scalable than previous approaches (Grathwohl et al., 2019; Yang & Ji, 2021; Yang et al., 2023). Additionally, we show that the denoising objective improves classification performance and robustness.

We motivate our approach through the perspectives of joint energy modeling and denoising score matching in Section 2. We then present our method in detail in Section 3, and evaluate it numerically in Section 4. We discuss our results in connection with the literature in Section 5. We conclude and evoke future research directions in Section 6.

## 2 JOINT ENERGY-SCORE MODELS

Consider a dataset of labeled images $(\mathbf{x}_i, c_i)_{1 \le i \le n}$ with images $\mathbf{x}_i \in \mathbb{R}^d$ and class labels $c_i \in \{1, \ldots, C\}$ of i.i.d. pairs sampled from a joint probability distribution $p(\mathbf{x}, c)$. Typical machine learning tasks then correspond to estimating conditional or marginal distributions: training a classifier amounts to learning a model of $p(c|\mathbf{x})$, while (conditional) generative modeling targets $p(\mathbf{x})$ or $p(\mathbf{x}|c)$. Rather than solving each of these problems separately, this paper argues for training a *single* model $p_\theta(\mathbf{x}, c)$ of the *joint* distribution $p(\mathbf{x}, c)$, following Hinton (2007); Grathwohl et al. (2019).

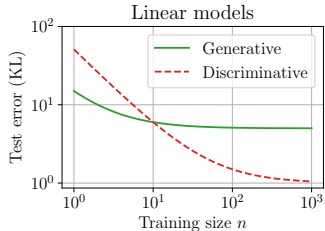 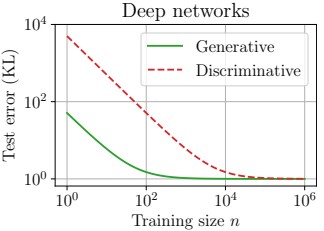

Figure 1: Illustration of the generalization bounds $b + \frac{v}{n}$ in two idealized settings with stylized values for $b$ and $v$. **Left:** bias-dominated setting with $b^{\text{gen}} = 5$, $b^{\text{dis}} = 1$, $v^{\text{gen}} = 20$, $v^{\text{dis}} = 100$. **Right:** variance-dominated setting with $b^{\text{gen}} = b^{\text{dis}} = 1$, $v^{\text{gen}} = 100$, $v^{\text{dis}} = 10000$.

The conditional and marginal distributions can then be recovered from the joint model with

$$p_\theta(c|\mathbf{x}) = \frac{p_\theta(\mathbf{x}, c)}{\sum_{c=1}^{C} p_\theta(\mathbf{x}, c)}, \qquad p_\theta(\mathbf{x}) = \sum_{c=1}^{C} p_\theta(\mathbf{x}, c), \qquad p_\theta(\mathbf{x}|c) = \frac{p_\theta(\mathbf{x}, c)}{p_\theta(c)}. \qquad (1)$$

We note that $p_\theta(c)$ is intractable to compute, but in the following we will only need the score of the conditional distribution $\nabla_\mathbf{x} \log p_\theta(\mathbf{x}|c)$, which is equal to $\nabla_\mathbf{x} \log p_\theta(\mathbf{x}, c)$ independently of $p_\theta(c)$.

In Section 2.1, we motivate the joint approach and detail its potential benefits over separate modeling of the conditional distributions. We then extend this to the context of diffusion models in Section 2.2, which leads to an interpretation of adversarial gradients as a difference of denoisers.

## 2.1 ADVANTAGES OF JOINT OVER CONDITIONAL MODELING

We first focus on classification. Models of the conditional distribution $p(c|\mathbf{x})$ derived from a model of the joint distribution $p(\mathbf{x}, c)$ are referred to as *generative* classifiers, as opposed to *discriminative* classifiers which directly model $p(c|\mathbf{x})$. The question of which approach is better goes back at least to Vapnik (1999). It was then generally believed that discriminative classifiers were better: quoting Vapnik (1999), "when solving a given problem, try to avoid solving a more general problem as an intermediate step". This has led the community to treat the modeling of $p(\mathbf{x})$ and $p(c|\mathbf{x})$ as two separate problems, though there were some joint approaches (Ng & Jordan, 2001; Raina et al., 2003; Ulusoy & Bishop, 2005; Lasserre et al., 2006; Ranzato et al., 2011). In the deep learning era, the joint approach was revitalized by Grathwohl et al. (2019) with several follow-up works (Liu & Abbeel, 2020; Grathwohl et al., 2021; Yang & Ji, 2021; Yang et al., 2023). They showed the many benefits of generative classifiers, such as better calibration and increased robustness to adversarial attacks. It was also recently shown in Jaini et al. (2024) that generative classifiers are much more human-aligned in terms of their errors, shape-vs-texture bias, and perception of visual illusions. We reconcile these apparently contradictory perspectives by updating the arguments in Ng & Jordan (2001) to the modern setting.

Consider that we have a parametrized family $\{p_\theta(\mathbf{x}, c), \theta \in \mathbb{R}^m\}$ (for instance, given by a neural network architecture). Generative and discriminative classifiers are respectively trained to maximize likelihood or minimize cross-entropy:

$$\theta_n^{\text{gen}} = \arg\max_\theta \frac{1}{n} \sum_{i=1}^{n} \log p_\theta(\mathbf{x}_i, c_i), \qquad \theta_n^{\text{dis}} = \arg\max_\theta \frac{1}{n} \sum_{i=1}^{n} \log p_\theta(c_i|\mathbf{x}_i). \qquad (2)$$

We measure the expected generalization error of the resulting classifiers in terms of the Kullback-Leibler divergence on a test point $\mathbf{x} \sim p(\mathbf{x})$:

$$\varepsilon(\theta) = \mathbb{E}_{\mathbf{x} \sim p(\mathbf{x})}[\text{KL}(p(c|\mathbf{x}) \,\|\, p_\theta(c|\mathbf{x}))]. \qquad (3)$$

We show that it can be described with "bias" and "variance" constants $b^{\text{gen}}, b^{\text{dis}}, v^{\text{gen}}, v^{\text{dis}}$:

$$\mathbb{E}[\varepsilon(\theta_n^{\text{gen}})] = b^{\text{gen}} + \frac{v^{\text{gen}}}{2n} + o\left(\frac{1}{n}\right), \qquad \mathbb{E}[\varepsilon(\theta_n^{\text{dis}})] = b^{\text{dis}} + \frac{v^{\text{dis}}}{2n} + o\left(\frac{1}{n}\right), \qquad (4)$$

where the expected values average over the randomness of the training set. This is derived with standard arguments in Appendix D, where the constants are explicited as functions of $p$ and $\{p_\theta\}$. The bias arises from model misspecification, while the variance measures the sample complexity of the model, as we detail below. We note that these results are only asymptotic, and the constants hidden in the little-$o$ notation could be exponential in the dimension. Nevertheless, these results provide evidence of the two distinct regimes found by Ng & Jordan (2001), as illustrated in Figure 1.

Which approach is better then depends on the respective values of the bias and variance constants. The bias is the asymptotic error when $n \to \infty$, and thus only depends on approximation properties. It always holds that $b^{\mathrm{dis}} \leq b^{\mathrm{gen}}$, as discriminative classifiers directly model $p(c|\mathbf{x})$ and thus do not pay the price of modeling errors on $p(\mathbf{x})$. The variance measures the number of samples needed to reach this asymptotic error. In the case of zero bias, we have $v^{\mathrm{gen}} \leq v^{\mathrm{dis}}$ (and further $v^{\mathrm{dis}} = m$ the number of parameters): generative classifiers learn faster (with fewer samples) as they exploit the extra information in $p(\mathbf{x})$ to estimate the parameters. With simple models with few parameters (e.g., linear models, as in Ng & Jordan (2001)), we can expect the bias to dominate, which has led the community to initially favor discriminative classifiers. On the other hand, with complex expressive models (e.g., deep neural networks) that have powerful inductive biases, we can expect the variance to dominate. In this case, generative classifiers have the advantage, as can be seen in the recent refocus of the community towards generative models and classifiers (Grathwohl et al., 2019; Jaini et al., 2024). The success of self-supervised learning also indicates the usefulness of modeling $p(\mathbf{x})$ to learn $p(c|\mathbf{x})$.

For similar reasons, a joint approach can also be expected to lead to benefits in conditional generative modeling. For instance, Dhariwal & Nichol (2021) found that conditional generative models could be improved by classifier guidance, and Ho & Salimans (2022) showed that these benefits could be more efficiently obtained with a single model.

## 2.2 Joint modeling with diffusion models

The joint approach offers significant statistical advantages, but also comes with computational challenges. Indeed, it requires to learn a model of a probability distribution over the high-dimensional image $\mathbf{x}$, while in a purely discriminative model it only appears as a conditioning variable (the probability distribution is over the discrete class label $c$). In Grathwohl et al. (2019), the generative model is trained directly with maximum-likelihood and Langevin-based MCMC sampling, which suffers from the curse of dimensionality and does not scale to large datasets such as ImageNet. We revisit their approach in the context of denoising diffusion models, which have empirically resolved these issues. We briefly introduce them here, and explain the connections with adversarial robustness in the context of generative classifiers.

**Diffusion models and denoising.** In high-dimensions, computing maximum-likelihood parameters is typically intractable due to the need to estimate normalizing constants. As a result, Hyvärinen proposed to replace maximum-likelihood with score matching (Hyvärinen & Dayan, 2005), which removes the need for normalizing constants and is thus computationally feasible. However, its sample complexity is typically much larger as a result of this relaxation (Koehler et al., 2022). Song et al. (2021) showed that this computational-statistical tradeoff can be resolved by considering a diffusion process (Sohl-Dickstein et al., 2015; Ho et al., 2020; Song & Ermon, 2019; Kadkhodaie & Simoncelli, 2021), where maximum-likelihood and score matching become equivalent.

Denoising diffusion models consider the joint distribution $p_\sigma(\mathbf{y}, c)$ of noisy images $\mathbf{y}$ together with class labels $c$ for every noise level $\sigma$. Formally, the noisy image $\mathbf{y}$ is obtained by adding Gaussian white noise of variance $\sigma^2$ to the clean image $\mathbf{x}$:

$$\mathbf{y} = \mathbf{x} + \sigma\boldsymbol{\epsilon}, \quad \boldsymbol{\epsilon} \sim \mathcal{N}(0, \mathrm{Id}). \tag{5}$$

Note that the class label $c$ becomes increasingly independent from $\mathbf{y}$ as $\sigma$ increases.

Modeling $p(\mathbf{y})$ or $p(\mathbf{y}|c)$ by score matching then amounts to performing unconditional or class-conditional denoising (Vincent, 2011), thanks to an identity attributed to Tweedie and Miyasawa (Robbins, 1956; Miyasawa et al., 1961; Raphan & Simoncelli, 2011):

$$\nabla_\mathbf{y} \log p(\mathbf{y}) = \frac{\mathbb{E}[\mathbf{x} \,|\, \mathbf{y}] - \mathbf{y}}{\sigma^2}, \qquad \nabla_\mathbf{y} \log p(\mathbf{y}|c) = \frac{\mathbb{E}[\mathbf{x} \,|\, \mathbf{y}, c] - \mathbf{y}}{\sigma^2}, \tag{6}$$

see e.g. Kadkhodaie & Simoncelli (2021) for proof. Note that $\mathbb{E}[\mathbf{x} \mid \mathbf{y}]$ and $\mathbb{E}[\mathbf{x} \mid \mathbf{y}, c]$ are the best approximations of $\mathbf{x}$ given $\mathbf{y}$ (and $c$) in mean-squared error, and are thus the optimal unconditional and conditional denoisers. A joint model $p_\theta(\mathbf{y}, c)$ thus gives us access to unconditional and class-conditional denoisers $\mathcal{D}_\mathrm{u}$ and $\mathcal{D}_\mathrm{c}$,

$$\mathcal{D}_\mathrm{u}(\mathbf{y}) = \mathbf{y} + \sigma^2 \nabla_{\mathbf{y}} \log p_\theta(\mathbf{y}), \qquad \mathcal{D}_\mathrm{c}(\mathbf{y}, c) = \mathbf{y} + \sigma^2 \nabla_{\mathbf{y}} \log p_\theta(\mathbf{y}|c), \tag{7}$$

which can be trained to minimize mean-squared error:

$$\min \mathbb{E}\Big[\|\mathbf{x} - \mathbf{y} - \sigma^2 \nabla_{\mathbf{y}} \log p_\theta(\mathbf{y})\|^2\Big], \qquad \min \mathbb{E}\Big[\|\mathbf{x} - \mathbf{y} - \sigma^2 \nabla_{\mathbf{y}} \log p_\theta(\mathbf{y}|c)\|^2\Big]. \tag{8}$$

**Adversarial gradients.** In this formulation, the adversarial classifier gradients (on noisy images) can be interpreted as a scaled difference of conditional and unconditional denoisers:

$$\nabla_{\mathbf{y}} \log p_\theta(c|\mathbf{y}) = \nabla_{\mathbf{y}} \log p_\theta(\mathbf{y}|c) - \nabla_{\mathbf{y}} \log p_\theta(\mathbf{y}) = \frac{1}{\sigma^2}(\mathcal{D}_\mathrm{c}(\mathbf{y}, c) - \mathcal{D}_\mathrm{u}(\mathbf{y})). \tag{9}$$

The adversarial gradient on clean images is obtained by sending the noise level $\sigma \to 0$, highlighting potential instabilities. Equation (9) provides a novel perspective on adversarial robustness (which was implicit in Ho & Salimans (2022)): adversarial gradients are the details in an infinitesimally noisy image that are recovered when the denoiser is provided with the class information. Being robust to attacks, which requires small adversarial gradients, thus requires the conditional denoiser to mostly ignore the provided class information when it is inconsistent with the input image at small noise levels. *The denoising objective (8) thus directly regularizes the adversarial gradients.* We discuss other connections between additive input noise and adversarial robustness in Section 5.1.

**Likelihood evaluation.** We also note that modeling the log-probability rather than the score has several advantages in diffusion models. First, it potentially allows to evaluate likelihoods in a single forward pass, as done in Choi et al. (2022); Yadin et al. (2024). Second, it leads to generative classifiers that can be evaluated much more efficiently than recent approaches based on score diffusion models (Li et al., 2023; Clark & Jaini, 2023; Jaini et al., 2024).

## 3 ARCHITECTURE AND TRAINING

The previous section motivated the joint approach and led to a unification between classification and denoising tasks. We now focus on these two problems, and describe our parameterization of the joint log-probability density (Section 3.1), the GradResNet architecture (Section 3.2), and the training procedure (Section 3.3).

### 3.1 PARAMETERIZATION OF JOINT LOG-PROBABILITY

We want to parameterize the joint log-probability density $\log p(c, \mathbf{y})$ over (noisy) images $\mathbf{y} \in \mathbb{R}^d$ and image classes $c$ using features computed by a neural network $f \colon \mathbb{R}^d \mapsto \mathbb{R}^K$. In classification, the class logits are parameterized as a linear function of the features

$$\log p_\theta(c|\mathbf{y}) = (\mathbf{W}f(\mathbf{y}))_c - \mathrm{LogSumExp}_{c'}\big((\mathbf{W}f(\mathbf{y}))_{c'}\big), \tag{10}$$

with $\mathbf{W}$ a matrix of size $C \times K$ where $C$ is the number of classes. We propose to parameterize the log-probability density of the noisy image distribution as a *quadratic* function of the features

$$\log p_\theta(\mathbf{y}) = -\frac{1}{2}(\mathbf{w}^\mathrm{T} f(\mathbf{y}))^2 - \log Z(\theta), \tag{11}$$

where $\mathbf{w} \in \mathbb{R}^K$ and $Z(\theta)$ is a normalizing constant that depends only on the parameters $\theta$. We thus have the following parameterization of the joint log-density

$$\log p_\theta(\mathbf{y}, c) = -\frac{1}{2}(\mathbf{w}^\mathrm{T} f(\mathbf{y}))^2 + (\mathbf{W}f(\mathbf{y}))_c - \mathrm{LogSumExp}_{c'}\big((\mathbf{W}f(\mathbf{y}))_{c'}\big) - \log Z(\theta). \tag{12}$$

Finally, the log-density of noisy images $\mathbf{y}$ conditioned on the class $c$ can be simply accessed with $\log p_\theta(\mathbf{y}|c) = \log p_\theta(\mathbf{y}, c) - \log p_\theta(c)$, where $\log p_\theta(c)$ is intractable but not needed in practice, as explained in Section 2. We chose the task heads to be as simple as possible, being respectively linear (for classification) and quadratic (for denoising) in the features. Preliminary experiments showed that additional complexity did not result in improved performance.

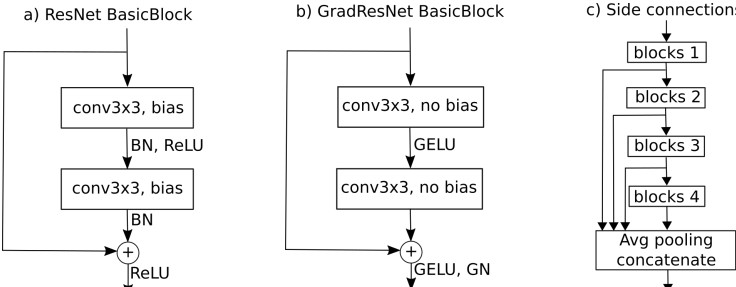

Figure 2: **Left:** ResNet BasicBlock with bias parameters, batch-normalization (BN) layers and ReLUs. **Middle:** GradResNet BasicBlock with bias-free convolutional layers, GELUs, and a single group-normalization (GN) layer. **Right:** Illustration of the proposed side connections.

## 3.2   GRADRESNET ARCHITECTURE

We now specify the architecture of the neural network $f_\theta$. As discussed in Section 2, realizing the potential of the joint modeling approach requires that the modeling bias is negligible, or in other words, that the inductive biases of the architecture and training algorithm are well-matched to the data distribution. Fortunately, we can rest on more than a decade of research on best architectures for image classification and denoising. Architectures for these two tasks are however quite different, and the main difficulty lies in unifying them: in particular, we want the computational graph of the *gradient* of a classifier network to be structurally similar to a denoising network.

In a recent work, Hurault et al. (2021) presented a gradient-based denoiser for Plug-and-Play image restoration, and they found that "directly modeling [the denoiser] as [the gradient of] a neural network (e.g., a standard network used for classification) leads to poor denoising performance". To remedy this problem, we thus propose the following architectural modifications on a ResNet18 backbone:

**Smooth activation function:** Cohen et al. (2021) first proposed to use the gradient of a feedforward neural network architecture as an image denoiser for Plug-and-Play image restoration. They write "an important design choice is that all activations are continuously differentiable and smooth functions". We thus replace ReLUs with smooth GELUs, following recent improvements to the ResNet architecture (Liu et al., 2022).

**Normalization:** Batch-normalization (Ioffe & Szegedy, 2015) is a powerful optimization technique, but its major drawback is that it has different behaviors in `train` and `eval` modes, *especially for the backward pass*. We hence replace batch-normalizations with group-normalizations (Wu & He, 2018), leading to a marginal decrease in classification performance on ImageNet (Wu & He, 2018). Moreover, we found that reducing the number of normalization layers is beneficial for denoising performance. While ResNets apply a normalization after each convolutional layer, we apply a group-normalization at the end of each basic block (see Figure 2).

**Bias removal:** Mohan et al. (2020) have shown that removing all bias parameters in CNN denoisers enable them to generalize across noise levels outside their training range. We decide to adopt this modification, removing all biases of convolutional, linear, and group-norm layers.

**Mimicking skip connections:** The computational graph of the gradient of a feedforward CNN can be viewed as a UNet, where the forward pass corresponds to the encoder (reducing image resolution), and the backward pass corresponds to the decoder (going back to the input domain). Zhang et al. (2021) show that integrating residual connections in the UNet architecture is beneficial for denoising performance. In order to emulate these residual connections in our gradient denoiser, we add side connections to our ResNet as illustrated in Figure 2 c).

With all these modification, we end up with a new architecture that we name *GradResNet*, which is still close to the original ResNet architecture. As we will show in the numerical experiments in Section 4, this architecture retains a strong accuracy and obtains competitive denoising results.

Table 1: CIFAR-10 experimental results. Training time is measured on a single NVIDIA A100 GPU, except for JEM, whose results and training time are taken from Grathwohl et al. (2019). Best results are highlighted in bold for each category of models.

| Task | Architecture | Accuracy (%) | PSNR$_{\sigma=15}$ | PSNR$_{\sigma=25}$ | PSNR$_{\sigma=50}$ | Training time (h) |
|---|---|---|---|---|---|---|
| *Classification* | ResNet18 | **96.5** | N.A. | N.A. | N.A. | **0.7** |
| | GradResNet | 96.1 | N.A. | N.A. | N.A. | **0.7** |
| *Denoising* | DnCNN | N.A. | 32.05 | 29.07 | 25.23 | **0.8** |
| | DRUNet | N.A. | 32.12 | 29.20 | **25.52** | 0.9 |
| | GradResNet | N.A. | **32.21** | **29.28** | 25.51 | 1.3 |
| *Joint* | JEM (WideResNet) | 92.9 | N.A. | N.A. | N.A. | 36 |
| | GradResNet | **96.3** | 31.90 | 29.00 | 25.25 | **1.3** |

## 3.3 Training

**Training objectives.** Our model of the joint log-distributions $\log p_\theta(\mathbf{y}, c)$ is the sum of two terms, $\log p_\theta(c|\mathbf{y})$ and $\log p_\theta(\mathbf{y})$. Our training objective is thus naturally a sum of two losses: a cross-entropy loss for the class logits $\log p_\theta(c|\mathbf{y})$, and a denoising score matching loss for $\log p_\theta(\mathbf{y})$. Both objectives are integrated over noise levels $\sigma$. For simplicity, we do not add any relative weighting of the classification and denoising objectives. Our final training loss is thus

$$\ell(\theta) = \mathbb{E}\Big[ -\log p_\theta(c|\mathbf{y}) + \big\| \sigma \nabla_\mathbf{y} \log p_\theta(\mathbf{y}) + \boldsymbol{\epsilon} \big\|^2 \Big], \tag{13}$$

where the expectation is over $(\mathbf{x}, c) \sim p(\mathbf{x}, c)$, $\boldsymbol{\epsilon} \sim \mathcal{N}(0, \mathrm{Id})$, and $\sigma \sim p(\sigma)$ (described below). The main advantage of the joint learning framework is that the two tasks (modeling $p(\mathbf{x})$ and $p(c|\mathbf{x})$) naturally combine in a single task (modeling $p(\mathbf{x}, c)$), thus removing the need for tuning a Lagrange multiplier trading off the two losses (where we interpret the denoising objective as a lower-bound on the negative-log-likelihood (Song et al., 2021)). Finally, we note that although redundant, one can also add a conditional denoising objective $\big\| \sigma \nabla_\mathbf{y} \log p_\theta(\mathbf{y}|c) + \boldsymbol{\epsilon} \big\|^2$, or use it to replace the unconditional denoising objective. We have empirically found no difference between these choices. Full experimental details can be found in Appendix A.

**Training distributions.** Data augmentation is used in both image classification and image denoising to reduce overfitting, but with different augmentation strategies. On the one hand, state of the art image classification models, e.g. Liu et al. (2022), are trained with the sophisticated MixUp (Zhang et al., 2017a) and CutMix (Yun et al., 2019) augmentations which significantly improve classification performance but introduce visual artifacts and thus lead to non-natural images. On the other hand, state-of-the-art denoisers like Restormer (Zamir et al., 2022) or Xformer (Zhang et al., 2023) use simple data-augmentation techniques like random image crops (without padding) and random horizontal flips, which leave the distribution of natural images unchanged. We thus use these different data-augmentation techniques for each objective, and thus the two terms in our objective are computed over different image distributions. Though this contradicts the assumptions behind the joint modeling approach, we found that the additional data-augmentations on the classification objective significantly boosted accuracy without hurting denoising performance. We also found beneficial to use different noise level distributions $p(\sigma)$ for the two tasks. In both cases, $\sigma$ is distributed as the square of a uniform distribution, with $\sigma_{\min} = 1$ and $\sigma_{\max} = 100$ for denoising but $\sigma_{\max} = 20$ for classification (relative to image pixel values in $[0, 255]$). On the ImageNet dataset, for computational efficiency, we set the image and batch sizes for the denoising objective according to a schedule following Zamir et al. (2022), explicited in Appendix A.

## 4 Numerical results

### 4.1 Classification and denoising via joint modeling

With the proposed joint modeling framework, we can perform classification and denoising at the same time. Training and implementation details are given in Section 3 and Appendix A. We measure the

Table 2: ImageNet experimental results. Training time is measured on a single NVIDIA A100 GPU.

| Task | Architecture | Accuracy (%) | PSNR$_{\sigma=15}$ | PSNR$_{\sigma=25}$ | PSNR$_{\sigma=50}$ | Training time (h) |
|---|---|---|---|---|---|---|
| *Classification* | ResNet18 | 69.8 | N.A. | N.A. | N.A. | 96 |
| *Joint* | GradResNet | 68.6 | 34.59 | 31.94 | 27.17 | 208 |

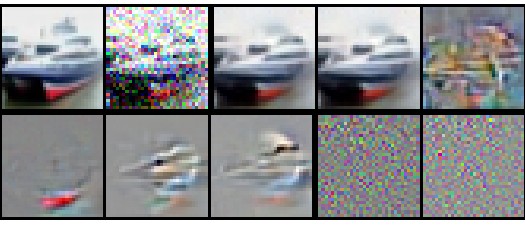

Figure 3: Denoising experiment. **Top, left-to-right:** Original CIFAR-10 test image, noisy image ($\sigma = 50$), denoised images with unconditional and conditional denoisers, and difference between them (magnified 500x). **Bottom, left-to-right:** Eigenvectors corresponding to the three largest (2.71,2.16, 2.03) and two lowest ($2.8 \times 10^{-5}, -1.9 \times 10^{-5}$) magnitude eigenvalues of the unconditional denoiser Jacobian. More examples are shown in Appendix C.

classification accuracy and the denoising PSNR averaged over the test set, compared with baselines. The classification baseline is a standard ResNet18 (He et al., 2016). As denoising baselines, we use two standard architectures, DnCNN (Zhang et al., 2017b) and DRUNet (Zhang et al., 2021). All baselines are trained with the same setup as our joint approach (but on a single objective).

Results are shown in Table 1 for CIFAR-10 (Krizhevsky et al., 2009) and in Table 2 for ImageNet (Russakovsky et al., 2015). We obtain classification and denoising performances that are competitive with the baselines. Importantly, our method provides a big computational advantage to the previous work JEM (Grathwohl et al., 2019), as can be seen from the training times in Table 1. JEM is based on maximum-likelihood training, which is very challenging to scale in high dimensions (as the authors of note, "training [...] can be quite unstable"), and therefore has been limited to $32 \times 32$-sized images, even in subsequent work (Yang et al., 2023). In contrast, denoising score matching is a straightforward regression, which leads to a stable and lightweight training that scales much more easily to ImageNet at full $224 \times 224$ resolution.

In Table 1, we report results for GradResNet trained on a single task to separate the impact of each objective. We remark that the denoising objective *improves* classification performance, confirming the arguments in Section 2.1. The classification objective however slightly degrades denoising performance. We also conduct several ablation experiments to validate our architecture choices in Table 3 in Appendix B. All our choices are beneficial for both classification and denoising performance, except for the removal for biases which very slightly affects denoising performance.

### 4.2 ANALYZING THE LEARNED DENOISERS

On the top row of Figure 3, we show an example of a noisy image denoised using the learned unconditional and class-conditional denoisers. We also show the difference between the two denoised images scaled by a factor of 500. The two denoised images look very similar, suggesting that the class conditioning is not very informative here.

Importantly, as we removed all the bias terms from the convolutional layers, our denoisers are bias-free. Mohan et al. (2020) have shown that bias-free deep denoisers $\mathcal{D}$ are locally linear operators, i.e., $\mathcal{D}(\mathbf{y}) = \nabla_{\mathbf{y}}\mathcal{D}(\mathbf{y})\mathbf{y}$, where $\nabla_{\mathbf{y}}\mathcal{D}(\mathbf{y})$ is the Jacobian of the denoiser evaluated at $\mathbf{y}$. To study the effect of a denoiser on a noisy image $\mathbf{y}$, one can compute the singular value decomposition of the denoiser's Jacobian evaluated at $\mathbf{y}$ (see Mohan et al. (2020) for details). The Jacobian of our unconditional denoiser is

$$\nabla_{\mathbf{y}}\mathcal{D}_{\mathrm{u}}(\mathbf{y}) = \mathrm{Id} + \sigma^2\nabla_{\mathbf{y}}^2 \log p_\theta(\mathbf{y}), \tag{14}$$

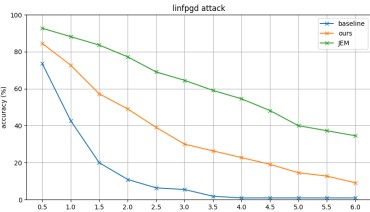 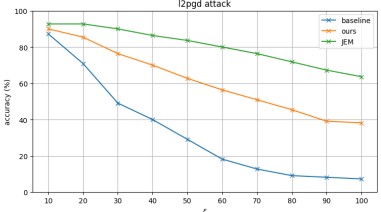

Figure 4: Adversarial attacks on CIFAR-10 test set. The baseline is a ResNet18 trained for classification only, ours is a GradResNet trained for classification and denoising, and JEM is the method proposed by Grathwohl et al. (2019). **Left:** $\ell^\infty$ PGD attack. **Right:** $\ell^2$ PGD attack.

where $\nabla^2$ denotes the Hessian. Since our denoiser is a gradient field, its Jacobian is symmetric, and can thus be diagonalized. The bottom row of Figure 3 shows the eigenvectors corresponding to the three largest and two lowest magnitude eigenvalues. Interestingly, eigenvectors corresponding to largest eigenvalues capture large scale features from the original image which are amplified by the denoiser. Conversely, lowest eigenvectors correspond to noisy images without shape nor structure that are discarded by the denoiser.

### 4.3 ADVERSARIAL ROBUSTNESS

In their seminal work on joint-modeling, Grathwohl et al. (2019) show that learning the joint distribution over images and classes leads to increased robustness to adversarial attacks. We perform projected gradient descent (PGD) adversarial attacks using the foolbox library (Rauber et al., 2020) on our baseline classifier and the classifier optimized for joint-modeling. Results are shown in Figure 4. We note that the proposed method increases adversarial robustness, but not as much as an MCMC-trained joint energy-based model (Grathwohl et al., 2019).

## 5 DISCUSSION AND RELATED WORK

### 5.1 ROBUST CLASSIFIERS

**Adversarial robustness.** Adversarial robustness is deeply related to noise addition. Indeed, Gu & Rigazio (2014) have shown that adding noise to adversarial attacks (referred to as *randomized smoothing*) can mitigate their effect. Further, Lecuyer et al. (2019) and follow-ups (Li et al., 2019; Cohen et al., 2019) showed that being robust to additive noise provably results in adversarial robustness using this strategy. This was recently shown to be practical in Maho et al. (2022). Note that noise addition *at test time* (as opposed to only during training) is critical to obtain robustness (Carlini & Wagner, 2017), an observation also made in Su & Kempe (2023). This appears in the instability of the $\sigma \to 0$ limit in eq. (9), and indicates that robustness tests as in Figure 4 should be computed over attack strength *and* noise levels.

Building classifiers that are robust to additive noise on the input image was considered in many works (note that this is separate from the issue of learning from noisy labels). Dodge & Karam (2017) found that fine-tuning standard networks on noisy images improve their robustness, but still falls short of human accuracy at large noise levels. A common benchmark (which also include corruptions beyond additive noise) was established in Hendrycks & Dietterich (2018). Effective approaches include carefully designing pooling operators (Li et al., 2020) and iterative distillation (Xie et al., 2020).

**Adversarial gradients and generative models.** Several works use denoisers or other generative models to improve adversarial or noise robustness, e.g., by denoising input images as a pre-processing step (Roy et al., 2018), or by leveraging an off-the-shelf generative model to detect and "purify" adversarial examples (Song et al., 2017). Grathwohl et al. (2019) unified the generative model and the classifier in a single model and showed that joint modeling results in increased adversarial robustness. Our work goes one step further by also making the connection to noise robustness and denoising through diffusion models, warranting further work in this direction.

## 5.2 SAMPLING AND ARCHITECTURE OF GRADIENT DENOISERS

**Gradient denoiser architectures.** Several works have studied the question of learning the score as a gradient. Most similar to our setup is Cohen et al. (2021), where the authors introduce the GraDnCNN architecture. When modeling directly the score with a neural network, Saremi (2019) show that the learned score is not a gradient field unless under very restrictive conditions. Nonetheless, Mohan et al. (2020) show that the learned score is approximately conservative (i.e., a gradient field), and Chao et al. (2023) introduce an additional training objective to penalize the non-conservativity. Finally, Horvat & Pfister (2024) demonstrate that the non-conservative component of the score is ignored during sampling with the backward diffusion, though they find that a conservative denoiser is necessary for other tasks such as dimensionality estimation. Our work provides several recommendations for the design of gradient-based denoisers.

**Sampling.** Preliminary experiments indicate that sampling from the distributions $p(\mathbf{x})$ and $p(\mathbf{x}|c)$ learned by our model, e.g. with DDPM (Ho et al., 2020), is not on par with standard diffusion models. Salimans & Ho (2021) noted that "specifying the score model by taking the gradient of an image classifier has so far not produced competitive results in image generation." While our changes to the ResNet architecture significantly alleviate these issues, its gradient might not have the right inductive biases to model the score function. Indeed, recent works (Salimans & Ho, 2021; Hurault et al., 2021; Yadin et al., 2024) rather model log-probabilities with (the squared norm of) a UNet, whose gradient is then more complicated (its computational graph is *not* itself a UNet). Understanding the inductive biases of these different approaches is an interesting direction for future work. After the completion of this work, we became aware of a related approach by (Guo et al., 2023). They tackle classification and denoising with a UNet network, and demonstrate competitive sampling results. This complements our conceptual arguments for the joint approach, confirms its potential, and calls for an explanation between the discrepancies in performance between the two architectures.

**Conditioning.** We also note that the class information is used differently in classification and conditional denoising. On the one hand, in classification, the class label is only used as an index in the computed logits (which allows for fast evaluation of all class logits). On the other hand, conditional denoisers typically embed the class label as a continuous vector, which is then used to modify gain and bias parameters in group-normalization layers (which is more expressive). This raises the question whether these two different approaches could be unified in the joint framework. An area where we expect the joint approach to lead to significant advances is in that of caption conditioning, where the classifier $\log p(c|\mathbf{y})$ is replaced by a captioning model. Training a *single* model on *both* objectives has the potential to improve the correspondence between generated images and the given caption. Another type of conditioning is with the noise level $\sigma$, also known as the time parameter $t$. In a recent work, Yadin et al. (2024) propose classification diffusion models (CDM). They model the joint distribution over images and discrete noise levels with denoising score matching and noise level classification. Both approaches are orthogonal and could be combined in future work.

## 6 CONCLUSION

In this paper, we have shown that several machine-learning tasks and issues are deeply connected: classification, denoising, generative modeling, adversarial robustness, and conditioning are unified by our joint modeling approach. Our method is significantly more efficient than previous maximum-likelihood approaches, and we have shown numerical experiments which give promising results on image (robust) classification and image denoising, though not yet in image generation.

We believe that exploring the bridges between these problems through the lens of joint modeling as advocated in this paper are fruitful research directions. For instance, can we understand the different inductive biases of ResNet and UNet architectures in the context of joint energy-based modeling? Can we leverage the connection between adversarial gradients and denoising to further improve classifier robustness? We also believe that the joint approach holds significant potential in the case where the class label $c$ is replaced by a text caption. Indeed, modeling the distribution of captions $p(c)$ is highly non-trivial and should be beneficial to conditional generative models for the same reasons than we used to motivate generative classifiers in Section 2.1. Finally, the joint approach yields a model of the log-probability $\log p(\mathbf{y})$, as opposed to the score $\nabla_{\mathbf{y}} \log p(\mathbf{y})$. We thus expect that this model could be used to empirically probe properties of high-dimensional image distributions.

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

## A  TRAINING AND IMPLEMENTATION DETAILS

**Experiments on CIFAR-10.**  We implement the proposed network and the training script using the PyTorch library (Paszke et al., 2019). We adapt the original ResNet18 (He et al., 2016) to $32 \times 32$ images by setting the kernel size of the first convolutional layer to 3 (originally 7) and the stride of the first two convolutional layers to 1. We also remove the max-pooling layer. To train our GradResNet, we use the AdamW optimizer with the default parameters except for the weight decay (0.05) and the learning rate ($3 \times 10^{-4}$). We use a cosine annealing schedule for the learning rate, without warm-up. We use a batch size of 64 for denoising and 128 for classification. The optimized loss is the sum of the cross-entropy loss and the mean-squared denoising objective (eq. 13). We run the optimization for 78k iterations, corresponding to 200 epochs for classification.

For classification, we use for data-augmentation random horizontal flips, padded random crops with padding of 4 pixels, and MixUp (Zhang et al., 2017a) and CutMix (Yun et al., 2019) as implemented in `torchvision.transforms` with the default parameters. We also add Gaussian white noise scaled by $\sigma$ that we draw from a squared uniform distribution in the range $[0, 20]$. For denoising, we simply use random flips. We add Gaussian noise with $\sigma \in [1, 100]$.

**Experiments on ImageNet.**  We use the standard ResNet18 architecture, but without the max-pooling layer. We set the learning rate to $2 \times 10^{-4}$ and weight decay to $10^{-4}$. We train the network for 400 epochs with a batch size of 512 for classification. We replace the random crop data augmentation with random resized crops of size $224 \times 224$. For denoising, we gradually change the image size and batch size across training, following Zamir et al. (2022). That is, we compute the denoising loss over image batches of size $64 \times 128 \times 128$ until epoch 150, then $40 \times 160 \times 160$ until epoch 260, then $32 \times 192 \times 192$ until epoch 340, and $16 \times 224 \times 224$ until epoch 400. Smaller images are extracted by taking random crops of the chosen size. Both objectives are computed with Gaussian noise with $\sigma$ the square of a uniform distribution supported in $[1, 70]$. All other hyperparameters are the same as for the CIFAR-10 dataset.

# B    ABLATION EXPERIMENTS

We present ablation experiments to validate our architecture choices of Section 3.2 in Table 3. Namely, we replace the GeLU non-linearity with the original ReLU, replace GroupNorm layers with the original BatchNorm, add biases to convolutional and GroupNorm layers, or remove side connections (see Figure 2).

Table 3: Ablation experiments on CIFAR-10 dataset.

| Task | Architecture | Accuracy (%) | PSNR$_{\sigma=15}$ | PSNR$_{\sigma=25}$ | PSNR$_{\sigma=50}$ |
|---|---|---|---|---|---|
| | GradResNet | **96.3** | 31.90 | 29.00 | 25.25 |
| | (ReLU) | 92.0 | 29.59 | 24.90 | 17.11 |
| *Joint* | (BatchNorm) | 11.9 | 24.33 | 20.84 | 18.06 |
| | (Biases) | 95.9 | **31.95** | **29.03** | **25.30** |
| | (No side co.) | 94.0 | 31.86 | 28.95 | 25.17 |

# C    DENOISING EXPERIMENTS

Additional denoising experiments are presented in Figure 5.

# D    BIAS-VARIANCE DECOMPOSITION OF THE GENERALIZATION ERROR

We sketch a derivation of the results. They can be rigorously proved under usual regularity conditions by a straightforward generalization of the proofs of asymptotic consistency, efficiency, and normality of the maximum-likelihood estimator (see, e.g., Hogg et al. (2013, Thereom 6.2.2)).

Given a function $\ell$, a probability distribution $p(\mathbf{x})$, and i.i.d. samples $\mathbf{x}_1, \ldots, \mathbf{x}_n$, consider the minimization problems

$$\theta_\star = \arg\min_\theta \; \mathbb{E}[\ell(\theta, \mathbf{x})], \qquad \theta_n = \arg\min_\theta \; \frac{1}{n}\sum_{i=1}^n \ell(\theta, \mathbf{x}_i). \tag{15}$$

We wish to estimate the fluctuations of the random parameters $\theta_n$ around the deterministic $\theta_\star$ when $n \to \infty$.

The estimator $\theta_n$ is defined by

$$\frac{1}{n}\sum_{i=1}^n \nabla_\theta \ell(\theta_n, \mathbf{x}_i) = 0. \tag{16}$$

A first-order Taylor expansion around $\theta_\star$ gives

$$\frac{1}{n}\sum_{i=1}^n \nabla_\theta \ell(\theta_\star, \mathbf{x}_i) + \frac{1}{n}\sum_{i=1}^n \nabla_\theta^2 \ell(\theta_\star, \mathbf{x}_i)(\theta_n - \theta_\star) = 0, \tag{17}$$

and thus

$$\theta_n = \theta_\star - \frac{1}{\sqrt{n}}\left(\frac{1}{n}\sum_{i=1}^n \nabla_\theta^2 \ell(\theta_\star, \mathbf{x}_i)\right)^{-1}\left(\frac{1}{\sqrt{n}}\sum_{i=1}^n \nabla_\theta \ell(\theta_\star, \mathbf{x}_i)\right). \tag{18}$$

By applying the law of large numbers to the first sum and the central limit theorem to the second sum (by definition of $\theta_\star$, $\mathbb{E}[\nabla_\theta \ell(\theta_\star, \mathbf{x})] = 0$), it follows that as $n \to \infty$ we have

$$\theta_n \sim \mathcal{N}\left(\theta_\star, \frac{1}{n}\Sigma\right), \tag{19}$$

with a covariance

$$\Sigma = \mathbb{E}\left[\nabla_\theta^2 \ell(\theta_\star, x)\right]^{-1} \mathrm{Cov}[\nabla_\theta \ell(\theta_\star, \mathbf{x})] \, \mathbb{E}\left[\nabla_\theta^2 \ell(\theta_\star, x)\right]^{-1}. \tag{20}$$

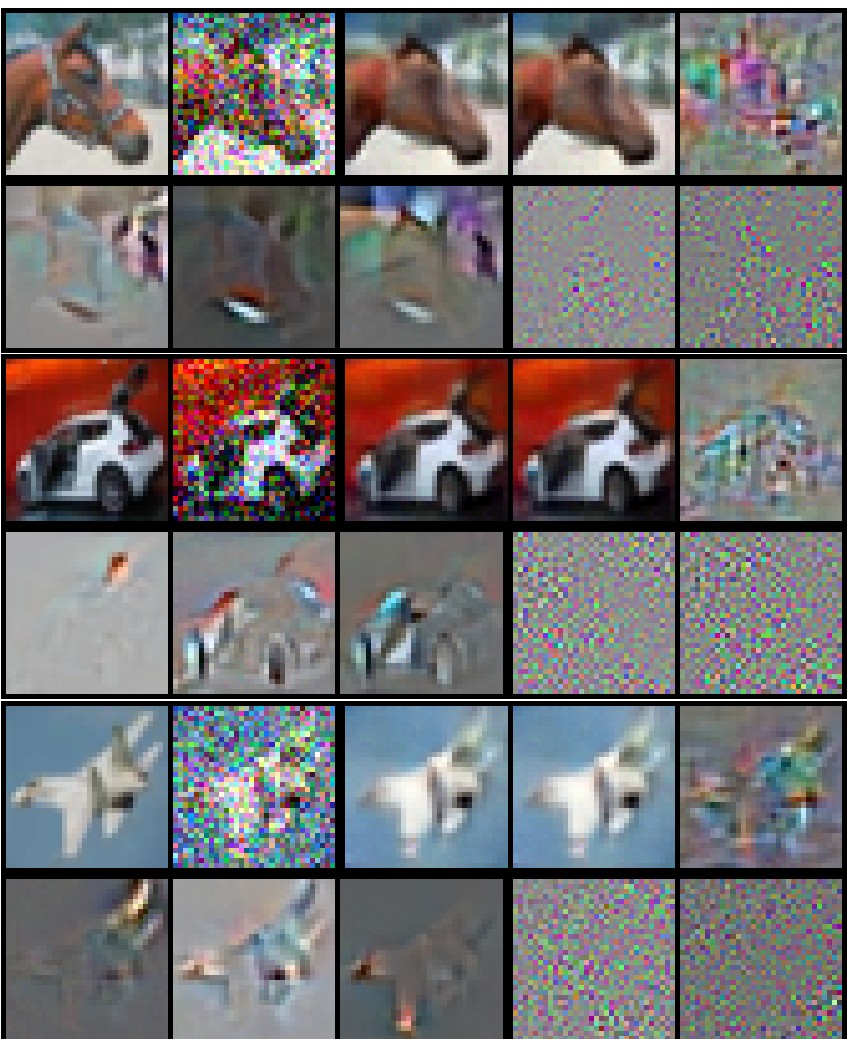

Figure 5: Additional denoising experiments with $\sigma = 50$. See Figure 3 for details.

For the generative and discriminative modeling tasks, we have

$$\theta_\star^{\text{gen}} = \arg\min_\theta \ \mathbb{E}[-\log p_\theta(\mathbf{x}, c)], \qquad \theta_n^{\text{gen}} = \arg\min_\theta \ \frac{1}{n}\sum_{i=1}^n -\log p_\theta(\mathbf{x}_i, c_i), \qquad (21)$$

$$\theta_\star^{\text{dis}} = \arg\min_\theta \ \mathbb{E}[-\log p_\theta(c|\mathbf{x})], \qquad \theta_n^{\text{dis}} = \arg\min_\theta \ \frac{1}{n}\sum_{i=1}^n -\log p_\theta(c_i|\mathbf{x}_i). \qquad (22)$$

In this setting, eq. (20) yields

$$\Sigma^{\text{gen}} = \mathbb{E}\Big[-\nabla_\theta^2 \log p_{\theta_\star^{\text{gen}}}(\mathbf{x}, c)\Big]^{-1} \text{Cov}\Big[-\nabla_\theta \log p_{\theta_\star^{\text{gen}}}(\mathbf{x}, c)\Big] \mathbb{E}\Big[-\nabla_\theta^2 \log p_{\theta_\star^{\text{gen}}}(\mathbf{x}, c)\Big]^{-1}, \qquad (23)$$

$$\Sigma^{\text{dis}} = \mathbb{E}\Big[-\nabla_\theta^2 \log p_{\theta_\star^{\text{dis}}}(c|\mathbf{x})\Big]^{-1} \text{Cov}\Big[-\nabla_\theta \log p_{\theta_\star^{\text{dis}}}(c|\mathbf{x})\Big] \mathbb{E}\Big[-\nabla_\theta^2 \log p_{\theta_\star^{\text{dis}}}(c|\mathbf{x})\Big]^{-1}. \qquad (24)$$

Note that we do not recover the Fisher information since expected values are with respect to the true distribution $p$, which is different from $p_{\theta_\star^{\text{gen}}}$ and $p_{\theta_\star^{\text{dis}}}$ under model misspecification.

These expressions can be plugged in a second-order Taylor expansion of the KL divergence: for any $\theta_n$ asymptotically normally distributed around $\theta_\star$ with covariance $\frac{1}{n}\Sigma$,

$$\mathbb{E}\big[\mathrm{KL}\big(p(c|\mathbf{x}) \,\big\|\, p_{\theta_n}(c|\mathbf{x})\big)\big] = \underbrace{\mathbb{E}\big[\mathrm{KL}\big(p(c|\mathbf{x}) \,\big\|\, p_{\theta_\star}(c|\mathbf{x})\big)\big]}_{b}$$

$$+ \frac{1}{2n}\underbrace{\mathrm{Tr}\Big(\Sigma\,\mathbb{E}\big[-\nabla_\theta^2 \log p_{\theta_\star}(c|\mathbf{x})\big]\Big)}_{v} + o\Big(\frac{1}{n}\Big). \qquad (25)$$

Finally, we obtain

$$b^{\mathrm{gen}} = \mathbb{E}\big[\mathrm{KL}\big(p(c|\mathbf{x}) \,\big\|\, p_{\theta_\star^{\mathrm{gen}}}(c|\mathbf{x})\big)\big], \qquad v^{\mathrm{gen}} = \mathrm{Tr}\Big(\Sigma^{\mathrm{gen}}\,\mathbb{E}\big[-\nabla_\theta^2 \log p_{\theta_\star^{\mathrm{gen}}}(c|\mathbf{x})\big]\Big), \qquad (26)$$

$$b^{\mathrm{dis}} = \mathbb{E}\Big[\mathrm{KL}\Big(p(c|\mathbf{x}) \,\big\|\, p_{\theta_\star^{\mathrm{dis}}}(c|\mathbf{x})\Big)\Big], \qquad v^{\mathrm{dis}} = \mathrm{Tr}\Big(\Sigma^{\mathrm{dis}}\,\mathbb{E}\big[-\nabla_\theta^2 \log p_{\theta_\star^{\mathrm{dis}}}(c|\mathbf{x})\big]\Big). \qquad (27)$$

By definition of $\theta_\star^{\mathrm{dis}}$, we have $b^{\mathrm{dis}} \le b^{\mathrm{gen}}$ (since maximizing likelihood is equivalent to minimizing KL divergence). The variance terms can be compared when the model is well-specified, i.e., there exists $\theta_\star$ such that $p = p_{\theta_\star}$. In this case, we have $\theta_\star^{\mathrm{gen}} = \theta_\star^{\mathrm{dis}} = \theta_\star$, so that $b^{\mathrm{gen}} = b^{\mathrm{dis}} = 0$, and the variances simplify to

$$v^{\mathrm{gen}} = \mathrm{Tr}\Big(\mathbb{E}\big[-\nabla_\theta^2 \log p_{\theta_\star}(\mathbf{x}, c)\big]^{-1} \mathbb{E}\big[-\nabla_\theta^2 \log p_{\theta_\star}(c|\mathbf{x})\big]\Big) \le m, \qquad (28)$$

$$v^{\mathrm{dis}} = \mathrm{Tr}(\mathrm{Id}) = m, \qquad (29)$$

using the Fisher information relationship

$$\mathbb{E}\big[-\nabla_\theta^2 \log p_{\theta_\star}(c|\mathbf{x})\big] = \mathrm{Cov}\big[-\nabla_\theta \log p_{\theta_\star}(c|\mathbf{x})\big], \qquad (30)$$

and the decomposition

$$\mathbb{E}\big[-\nabla_\theta^2 \log p_{\theta_\star}(\mathbf{x}, c)\big] = \mathbb{E}\big[-\nabla_\theta^2 \log p_{\theta_\star}(c|\mathbf{x})\big] + \underbrace{\mathbb{E}\big[-\nabla_\theta^2 \log p_{\theta_\star}(\mathbf{x})\big]}_{\succcurlyeq 0}. \qquad (31)$$

We thus have $v^{\mathrm{gen}} \le v^{\mathrm{dis}}$ in the well-specified case.

