# OpenReview forum: "Classification-denoising networks"
_ICLR.cc/2025/Conference — Submitted to ICLR 2025_

### Official Review · Reviewer_y1bd · 2024-10-29

**Soundness:** 3
**Presentation:** 3
**Contribution:** 2
**Rating:** 5
**Confidence:** 3

**Summary:**

This paper unifies the tasks of image classification and denoising through a joint probabilistic model of (noisy) images and class labels to address complementary issues, such as insufficient robustness and partial neglect of conditional information in each task. The paper first introduces a framework that uses a single network to parameterize the joint distribution \( p(y, c) \) for performing classification, class-conditional, and unconditional denoising. Then, an architecture is proposed to parameterize the joint log-probability density of images and labels.

**Strengths:**

1. The derived discrepancy in the denoiser complements the previously connections between adversarial robustness and denoising.
2. The proposed architecture combines inductive biases suitable for both denoising and classification.

**Weaknesses:**

1. In joint training, the performance of both classification and denoising is inferior to existing standalone classification or denoising methods.
2. Although joint training improves classification performance compared to separate training, the improvement is very limited and still falls short of ResNet18’s classification performance.
3. The authors claim that the proposed method can be applied to out-of-distribution detection, but no experiments were conducted to verify the effectiveness of the method in this regard.

**Questions:**

See Weaknesses.

---

> ### Author Response · Authors · 2024-11-22
>
> We thank the reviewer for their review.
>
> The reviewer is correct that our classification performance falls slightly below that of ResNet18. However, our GradResNet is significantly more robust than ResNet to adversarial attacks. It is generally expected that any form of robustness comes at the cost of (a little) classification accuracy. We believe GradResNet to be a valuable point on the Pareto frontier between classification performance, denoising performance, robustness, and training efficiency, when compared to standalone classification or denoising methods as well as JEM.

---

### Official Review · Reviewer_7aef · 2024-11-02

**Soundness:** 2
**Presentation:** 3
**Contribution:** 2
**Rating:** 5
**Confidence:** 2

**Summary:**

This paper integrates image denoising and image classification tasks, enabling the model to achieve enhanced robustness through joint learning. The architecture is based on ResNet-18, with modifications to improve performance by directly modeling the denoiser as the gradient of the neural network. The training objective combines cross-entropy loss with denoising score matching loss. Experimental results demonstrate that this approach outperforms previous jointly learned models and exhibits strong resilience to adversarial noise, along with a thorough analysis of its connection to score-based models.

**Strengths:**

+ Research on the joint learning of denoising and classification is limited; this paper highlights the potential of such integrated approaches.

+ The numerical results for classification surpass those of previous joint methods.

+ The relationship between the proposed method and the adversarial noise and energy-based models is analyzed in depth and discussed in detail.

**Weaknesses:**

- While this paper demonstrates superior performance in joint methods compared to others, it does not show significant advantages in standalone classification or denoising tasks. The denoising task under Gaussian noise can be viewed as a sampling process within a score-based diffusion model that uses Gaussian noise as a prior. In other words, the method presented in this paper sacrifices generative capabilities but learns a fixed noise-level denoiser in favor of enhanced classification performance. As a result, it outperforms JEM in classification but lacks the sampling ability of diffusion models and exhibits weaker resistance to adversarial noise compared to JEM.

**Questions:**

- Regarding the SVD decomposition of the Jacobian matrix, as described in line 427, it seems that any bias-free denoiser satisfies the condition \( D = \nabla_Y D Y \). Therefore, as long as a denoiser is effective and bias-free, the Jacobian will be decomposed to image feature information. I'm not sure what the significance of this experiment is, especially since Mohan et al. (2020) have already validated this.

-As mentioned in line 506, it seems that Guo et al. (2023) have already conducted a very similar study.


[1] Sreyas Mohan, Zahra Kadkhodaie, Eero P Simoncelli, and Carlos Fernandez-Granda. Robust and interpretable blind image denoising via bias-free convolutional neural networks. In International Conferenece on Learning Representations (ICLR), Addis Ababa, Ethiopia, April 2020.

[2] Qiushan Guo, Chuofan Ma, Yi Jiang, Zehuan Yuan, Yizhou Yu, and Ping Luo. Egc: Image generation and classification via a diffusion energy-based model. In Proceedings of the IEEE/CVF International Conference on Computer Vision, pp. 22952–22962, 2023.

---

> ### Author Response · Authors · 2024-11-22
>
> We thank the reviewer for their review.
>
> The reviewer is correct that our approach lacks sampling ability and is less robust than JEM to adversarial attacks. However, we wish to emphasize that our approach represents a significant gain in efficiency by more than 25x, and therefore is much more scalable to higher resolutions.
>
> The purpose of the Jacobian experiment in Figure 3 is simply to reproduce the observations of Mohan et al. (2020) in our setting, where we directly learn the log probability as opposed to the score (or the denoising function). In addition to demonstrating that we can learn similar adaptive filters, it represents a conceptual simplification, since our denoiser Jacobian is guaranteed to be symmetric as it the Hessian of a scalar function. This simplifies Jacobian analysis into a single orthogonal eigenbasis, as opposed to two orthogonal bases of left and right singular vectors. We will clarify this in the text.

---

### Official Review · Reviewer_oaVT · 2024-11-03

**Soundness:** 3
**Presentation:** 3
**Contribution:** 1
**Rating:** 1
**Confidence:** 5

**Summary:**

The paper propose a novel architecture that perform joint denoising and classification. It relies on tools from diffusion models such that the gradient of the denoiser is used to preform classification. The proposed algorithm and architecture are interesting and the mathematical formulation is sound. The authors perform experiments comparing to existing classifiers and denoisers showing on par performance.
Yet, the authors ignore a large body of works that already use diffusion models to perform classification and bear great similarities to the proposed work. I will detail this below.

**Strengths:**

The authors propose a nice joint formulation for denoising and classification. The mathematical derivation is clear and the formulation is sound. If there were not many prior works that did very similar things I would have recommended accepting the paper. But a proper work should compare to prior works...

**Weaknesses:**

The paper elegantly (or may be not) ignores all the prior works that use diffusion models to perform classification. In fact, they cite one work (your diffusion model is secretly one shot classifier) but don't compare to it. Indeed, the work they already cite, deals less with robustness but there are many other works that perform joint classification and denoising and study robustness. For example:
(CERTIFIED!!) ADVERSARIAL ROBUSTNESS FOR FREE!, ICLR 2023
Robust Classification via a Single Diffusion Model, ICML 2024
Diffusion Models are Certifiably Robust Classifiers, NeurIPS 2024

Mentioning these works, explaining the difference and comparing to them (!!!) is a must. Right now this is the main problem with the paper and by ignoring the existing prior art it is a clear reject.

**Questions:**

Why did you ignore existing works?

---

> ### Author Response · Authors · 2024-11-22
>
> We thank the reviewer for their review.
>
> We thank the reviewer for pointing out some references that we had missed. We will include them in the related work discussion. However, we think the reviewer's score is unjustified.
> - The reviewer suggests that we purposefully ignored part of the literature to avoid comparing to it. We have done our best attempt to cite all relevant works, which span several research areas: image classification, discriminative vs generative approaches, (joint) energy-based models, adversarial robustness, gradient-based denoising, and diffusion models, but we are not immune to oversights.
> - Regarding classification with diffusion models, in addition to Li et al. (2023) mentioned by the reviewer, we also cite Clark & Jaini (2023) and Jaini et al. (2024). We do not compare to these works, nor would it be meaningful to compare to the works mentioned by the reviewer, as none of them are energy-based models, which are the focus of our paper.
> - The ICLR concurrent work policy states that "if a paper was published (i.e., at a peer-reviewed venue) on or after July 1, 2024, authors are not required to compare their own work to that paper." Two of the papers mentioned by the reviewer are in that category. This even includes a paper published at a conference that has not taken place yet!

---

> > ### Comment · Reviewer_oaVT · 2024-11-24
> >
> > Thank you for your response but I don't really see any answer to any of my comments.

---

### Official Review · Reviewer_jHcX · 2024-11-07

**Soundness:** 2
**Presentation:** 2
**Contribution:** 2
**Rating:** 5
**Confidence:** 4

**Summary:**

This work proposes a joint framework for classifying and denoising and also proposes a new network called GradResNet. This single network was trained for both denoising and classification tasks, yielding comparable results to the prior works such as ResNet, DnCNN and so on.

**Strengths:**

- The attempt to jointly classify and denoise looks interesting. One network can be effectively used for both classification and denoising.

**Weaknesses:**

- It is less convincing why one needs to combine the tasks of classification and denoising into a single network. Moreover, it is unclear what are the novel contributions of this work over prior works such as JEM.
- The evaluation look simplified by using too small networks for the problem like ImageNet classification, by denoising too small images with simple synthetic noises, and by  using too limited benchmarks.
- Diffusion models were mentioned in this work, but there is no experimental results about them.

**Questions:**

- Please see the weakness section.

---

> ### Author Response · Authors · 2024-11-22
>
> We thank the reviewer for their review. We address their points below.
>
> ### "It is less convincing why one needs to combine the tasks of classification and denoising into a single network."
>
> The motivation of combining both tasks is the subject of Section 2. Because they both target complimentary aspects of the joint probability distribution $p(x,y)$, learning both with a single well-specified model provably leads to reduced variance and therefore can only improve performance in both tasks (see lines 168-180).
>
> ### "Moreover, it is unclear what are the novel contributions of this work over prior works such as JEM."
>
> The main contribution with respect to JEM is the greatly reduced training time (see last column of Table 1). This enables to scale to high-resolution images (we report results on ImageNet at 224x224 resolution, while JEM was limited to CIFAR-10 at 32x32 resolution).
>
> ### "The evaluation look simplified by using too small networks for the problem like ImageNet classification, by denoising too small images with simple synthetic noises, and by using too limited benchmarks."
>
> We don't understand the reviewer's point. The networks are based on ResNet-18, which seem to us to be an appropriate size for ImageNet. How can 224x224 images be too small? This is already 50 times more pixels than JEM. What does the reviewer mean by "simple synthetic noise"? Almost all diffusion models in the literature are based on Gaussian noise.
>
> ### "Diffusion models were mentioned in this work, but there is no experimental results about them."
>
> As mentioned at the beginning of Section 3, we decided to focus on classification and denoising in this work, and leave image generation to future work.
>
> We respectfully disagree with the reviewer that our contributions are poor. Scaling energy-based models to higher-resolution images has been a long-standing challenge in the literature. Although we do not report generation results, we believe our framework has great potential to improve scalability in EBMs and therefore to be of value to the community.

---

### Meta-Review · Area_Chair_rYM2 · 2024-12-20

**Metareview:**

Reviewers agree that the paper should not be accepted in its current form, due to underperformance wrt. prior work.
Authors rebuttal did not convince the reviewers of the papers limitation. Therefore the paper is recommended for rejection.

**Additional Comments On Reviewer Discussion:**

Authors did a poor job at the rebuttal. reviewers discussed that the need more explanations and authors did not respond anymore.

---

### Decision · Program_Chairs · 2025-01-22

Reject